# Pitfalls in the Diagnosis and Management of Hypercortisolism (Cushing Syndrome) in Humans; A Review of the Laboratory Medicine Perspective

**DOI:** 10.3390/diagnostics13081415

**Published:** 2023-04-14

**Authors:** Kade C. Flowers, Kate E. Shipman

**Affiliations:** 1Department of Clinical Chemistry, University Hospitals Sussex NHS Trust, Worthing BN11 2DH, UK; 2Department of Medical Education, Brighton and Sussex Medical School, University of Sussex, Falmer Campus, Brighton BN1 9PX, UK

**Keywords:** hypercortisolism, Cushing syndrome, analytical, spurious, mass spectrometry, dexamethasone, immunoassay

## Abstract

Biochemical confirmation of a diagnosis of hypercortisolism (Cushing syndrome) is vital to direct further investigations, especially given the overlap with non-autonomous conditions, such as pseudo-Cushing, and the morbidity associated with missed diagnoses. A limited narrative review was performed focusing on the laboratory perspective of the pitfalls of making a biochemical diagnosis of hypercortisolism in those presenting with presumed Cushing syndrome. Although analytically less specific, immunoassays remain cheap, quick, and reliable in most situations. Understanding cortisol metabolism can help with patient preparation, specimen selection (e.g., consideration of urine or saliva for those with possible elevations of cortisol binding globulin concentration), and method selection (e.g., mass spectrometry if there is a high risk of abnormal metabolites). Although more specific methods may be less sensitive, this can be managed. The reduction in cost and increasing ease of use makes techniques such as urine steroid profiles and salivary cortisone of interest in future pathway development. In conclusion, the limitations of current assays, particularly if well understood, do not impede diagnosis in most cases. However, in complex or borderline cases, there are other techniques to consider to aid in the confirmation of hypercortisolism.

## 1. Introduction

Hypercortisolism (the physical manifestation of which can be known as Cushing syndrome) is rare. However, it provides a subject for many excellent reviews and expert consensus statements due to the management complexity [1]. Hypercortisolaemia can occur in the absence of classical symptoms or a specific cause that requires intervention, e.g., hypercortisolaemia in alcohol abuse or depression [1]. Therefore, like all conditions, the performance of the diagnostic tests is greatly improved by careful patient selection, with disease prevalence directly impacting the positive and negative predictive power of the selected tests. This is particularly vital where the diagnostic tests have imperfect sensitivity and/or specificity [1,2].

This invited limited narrative review aims to summarise some of the main issues with the commonly used modern laboratory tests employed to manage humans presenting with symptoms of hypercortisolaemia. It is not to replace clinical practice guidelines based on systematic reviews and expert consensus [1]. Screening of single symptoms, e.g., hypertension, is out of the scope of the review and the focus is on the laboratory perspective. Medline was the primary search engine utilised (December 2022 to February 2023), with citations and reference lists being used, as well as the expected search terms, for example, ‘Cushing’, ‘hypercortisolaemia’, etc. 

Principal conclusions are that standard assays and pathways are appropriate in most cases of possible Cushing syndrome. However, there are many pre-analytical and analytical challenges; therefore, there should be a low threshold to discuss alternative testing strategies with local experts if results are not consistent with presentation. Improving technologies and reduction in cost may influence testing strategies in the future, particularly the role of salivary cortisone and urine steroid profiles.

## 2. Clinical Syndrome and Steroid Metabolism

### 2.1. Clinical Syndrome

In order to understand the analytical issues of confirming hypercortisolaemia in apparent hypercortisolism, it is important to appreciate that the clinical diagnosis of hypercortisolism is difficult due to an array of non-specific symptoms, e.g., central obesity, hypertension [1]. Testing should primarily be limited, i.e., not performed in everyone with simple obesity or hypertension, and staged to confirm the diagnosis and then the cause investigated [1]. As exogenous hypercortisolaemia is common, careful history and examination is required to prevent over-testing and over-diagnosis, but also, importantly, it is required due to the cross-reactivity of exogenous steroids in cortisol assays, which increase the risk of spurious results (see discussions below). Figure 1 presents the various conditions that can result in hypercortisolism.

In cyclical Cushing syndrome, symptoms can fluctuate or be permanent, but cortisol concentration will vary. Therefore, serial testing is required to detect hypercortisolaemia [3]. Exogenous hypercortisolism may appear to be cyclical-dependent on intake, and it is not only limited to those with known glucocorticoid intake. For example, unintentional exogenous exposure can occur due to potent glucocorticoids hidden in cosmetics and herbal creams [4,5,6]. Pseudo-Cushing syndrome is defined as temporary or permanent hypercortisolism caused by either appropriate physiological stimuli, e.g., major surgery, or conditions such as obesity, diabetes mellitus, and chronic alcoholism [7,8]. 

### 2.2. Steroid Metabolism

Cortisol acts on all cells via the intracellular glucocorticoid receptor (GR). The control and metabolism will be briefly discussed below.

#### 2.2.1. Homeostasis and Diurnal Variation

Physiological cortisol secretion from the adrenal glands depends upon the hypothalamic corticotrophin releasing hormone (CRH) stimulating pituitary-derived adrenocorticotrophic hormone (ACTH, also known as corticotrophin) secretion, which in turn stimulates the adrenal glands to synthesise and secrete cortisol. There is marked diurnal variation (peak in early morning, nadir just after going to sleep, e.g., midnight) of cortisol driven by phasic and incremental increases in ACTH concentration [9,10,11]. The diurnal variation becomes evident at some point after birth, but it is likely present in all infants from 9 months old [12]. For further discussion, please see the specimen timing section below. CRH is a small peptide, is present in small quantities, and is not routinely measured as it is not released into the systemic circulation, unlike ACTH. 

#### 2.2.2. Steroid Synthesis and Interference

Cortisol is synthesised in the adrenal cortex from cholesterol in the adrenal zona fasciculata (see Figure 2). If synthesis is occurring normally, the concentrations of the precursors should be low. However, in pathological synthesis, precursors can accumulate which can cross-react with measurement techniques and in vivo steroid receptors.

#### 2.2.3. Cortisol Binding Globulin and Interference

Cortisol, like many other hormones, is largely bound in the circulation. Approximately 90% is bound to cortisol-binding globulin (CBG), 5% to other proteins, and only 5% free [16,17]. Only free cortisol is filtered in urine or saliva; therefore, only blood samples will allow the measurement of total cortisol concentration (see Figure 3). Total cortisol assays are affected by changes in the concentration of CBG [18]. For example, CBG increases in response to exogenous oral (but not transcutaneous) oestrogens (which should be stopped 6 weeks prior to testing) and decreases in the nephrotic syndrome (see Table 1) [1,19,20,21]. Although cortisol secretion is increased in stress, if protein concentrations are low, e.g., hypoalbuminemia and low CBG concentration, then a spuriously lower total cortisol concentration may be recorded [22]. 

If plasma total cortisol concentrations are high when urine concentrations are low or normal (or vice versa), there remains the possibility (if specific assay interference has been excluded) that there is an abnormality in CBG [27,29]. 

#### 2.2.4. Steroid Metabolism and Interference

Cortisol is glucuronidated by the liver, undergoing further metabolism in the kidneys [30]. This generates a large number of low concentration metabolites (see Figure 4). For an informative review, please see Krone et al., 2010 [31].

Cortisol is also metabolised to the inactive cortisone by the 11-hydroxysteroid dehydrogenase type 2 (11β-HSD2) found in salivary glands, the kidney, and hair [32]. This means the 5:1 ratio of cortisol:cortisone in serum (ratio demonstrates diurnal variation and significant intraindividual variation) is reversed to 1:5 in saliva [33,34,35]. Liquorice and chewing tobacco contain glycyrrhizic acid which inhibits this enzyme, resulting in less deactivation of cortisol to cortisone and increased GR stimulation [36]. For example, liquorice has been shown to increase salivary cortisol [37]. Other inhibitors of 11β-HSD2 include carbenoxolone and grapefruit juice, as well as gossypol (from cotton plants), phthalates (plasticisers), organotins (found in paint), alkylphenols, and perfluorinated substances (both commonly used as raw materials in manufacturing), but the clinical impact of these is unclear as cases are not widely reported [38,39]. 

**Figure 4 diagnostics-13-01415-f004:**
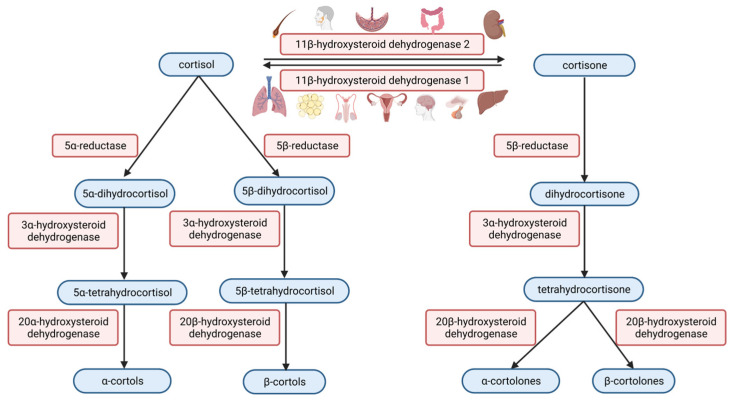
Cortisol metabolism pathway. The principal steps of cortisol metabolism are demonstrated. However, not all enzymes are found in all tissues, e.g., 11-hydroxysteroid dehydrogenase type 2 is found in hair, the salivary gland, kidney, colon, and placenta, and type 1 is found in the liver, lung, gonads, pituitary, brain, and adipose tissue [40]. This allows cortisol activity to be regulated at the tissue level. Created with Biorender.com [14,40].

## 3. Cortisol Analysis

Once history and examination confirm the clinical phenotype of Cushing syndrome, the first step of management is to obtain evidence of spontaneous hypercortisolism and exclude differentials, including pseudo-Cushing syndrome. In addition to diagnosis and identification of the cause, cortisol assessment can be key in the monitoring of therapy and assessment of remission. 

There are significant molecular issues with accurate detection and quantification of cortisol. Firstly, cortisol is a small molecule at an average of 0.36 kDa, whereas albumin is 66.5 kDa (almost 200 times larger). Secondly, cortisol circulates at low concentrations. For example, compare a concentration of cortisol of 140 nmol/L with a sodium concentration of 140 mmol/L (or 140,000,000 nmol/L), which is a 1,000,000-fold lower concentration. Thirdly, cortisol has many similarly structured precursors, metabolites, medications, and other steroid hormones, which can be present in the specimen and further complicate analysis due to cross-reactivity, despite method improvements [18]. There is a certified reference serum solution of a specified concentration for cortisol, but there is not one for cortisone. There is also a reference method (isotope dilution liquid chromatography/mass spectrometry or liquid chromatography/tandem mass spectrometry) for serum cortisol which improves traceability across manufacturer and assay types [34,41]. 

In the assumption that cortisol has been accurately quantified, then it is possible for those with Cushing syndrome to have normal results [42]. The following section will review the different analytical techniques and specimen requirements used to attempt to accurately quantify cortisol. It is important to note that studies which compare urine and salivary cortisol may also be comparing different analytical techniques. Therefore, when using a published diagnostic threshold, it is also important to note which assay was used to obtain the values, the patient preparation, and the timing of the specimen. 

### 3.1. Immunoassay

The use of labelled antibodies to analyse hormones at low concentrations revolutionised the management of endocrine disorders. Immunoassays continue to provide a cheap and automatable method for high throughput analysis using small specimen volumes (and different specimen types including hair and saliva), and they do not require specialist equipment or staff training. The immunoassay generates a single numerical concentration, and due to cross-reactivity (e.g., with cortisone and exogenous steroids), they can have increased sensitivity over the more analytically specific structural assays, albeit at the cost of specificity [1,18,27,43,44,45]. The positive bias of immunoassay, approximately 2.7-fold compared with mass spectrometry, may also be due to poor calibration [34,46]. However, immunoassay remains a robust detection method for hypercortisolaemia [47]. 

There are generic issues with immunoassay which should also be considered. Interference with other antibodies in blood, but not urine or saliva, is a variable problem which can be difficult to identify unless different samples and testing strategies have been employed [24,27,48]. The prozone (or hook) effect describes the phenomenon where the concentration of an analyte is so high that it exceeds the concentration of the antibody to which it must bind (see Figure 5) [24]. In ’one step’ sandwich immunoassays, the analyte must bind a capture antibody and the detection antibody prior to a wash step. However, if the binding sites are all occupied and the antigen is still in excess, then the detection antibodies will not necessarily be bound to the captured antigens; therefore, the signal is washed away in the wash step (increasing proportionally as antigen excess increases), reducing the reported concentration [24,27]. In ’two step’ immunoassays, there is an extra wash step before the addition of the detection antibody, which washes away all excess antigen that is not bound to the capture antibody, preventing the detection antibody from binding free antigen. Therefore, the hook effect does not occur. Despite this, the majority of automated immunoassays are ’one step’ and are therefore prone to the hook effect. Biotin interference is a problem for assays designed using streptavidin as a capture molecule but can be identified by asking patients about biotin intake and excluded by asking patients to refrain (although there is no consensus in regard to how long) [24]. Interference with the measurement label, e.g., ruthenium antibodies, can also occur [24].

### 3.2. Structural Assays: Mass Spectrometry and High-Performance Liquid Chromatography

Structural assays, such as mass spectrometry (MS) and high-performance liquid chromatography (HPLC) with detection, are less prone to cross-reactivity with exogenous steroids and endogenous metabolites due to specimen preparation, including separation steps, e.g., chromatography [1]. These methods can report the concentration of a range of steroids. Therefore, not only are MS and HPLC methods more specific to cortisol, but they can also detect abnormal concentrations, or ratios, of metabolites that cannot be differentiated by the purportedly cortisol-only immunoassays. As a consequence, in addition to confirming hypercortisolaemia, they can also indicate the cause, such as cancer, exogenous steroids, or inborn errors of metabolism [30,31]. However, the equipment is costly, the methods are more manual, and higher skill levels of laboratory staff are needed to process and interpret the results. The analytical specificity may also reduce the clinical sensitivity; therefore, lower thresholds for clinical suspicion may need to be used to account for the lack of cross-reactivity with abnormal concentrations of metabolites [30].

#### 3.2.1. Mass Spectrometry

Mass spectrometry is a detection and quantification technique that is more specific than immunoassay, as molecular species can be separated before analysis, as well as altered during analysis, e.g., to form parent and daughter ions to give more unique signals [18]. However, the data, being more complex than a simple cortisol concentration, may require specialist interpretation, and the equipment is not universally available to clinical laboratories [31,49]. As mass spectrometry distinguishes compounds of different masses, isobaric compounds can be indistinguishable, e.g., 21 deoxycortisol, 11 deoxycortisol, and corticosterone, hence the need for a chromatography step to separate these before MS detection [49]. Method design can prevent interference from exogenous glucocorticoids, such as prednisolone, and can be used when patients are on metyrapone [49]. 

Liquid chromatography tandem mass spectrometry (LC-MS/MS) is increasingly used in clinical laboratory services as machines are acquired, methods have been simplified, sped up, and automated, and small sample volumes can be analysed [49,50]. Gas chromatography–mass spectrometry (GC-MS), a precursor to LC-MS/MS, mainly remains a research or highly specialist technique. Considerable sample preparation is required for GC-MS, but this is at the benefit of making all cortisol metabolites and precursors detectable [31]. With LC-MS/MS, the analytes of choice are selected prior to analysis; therefore, unexpected conditions may be missed as not all analytes will necessarily be detected in comparison to GC-MS [31].

All protein must be removed prior to specimen analysis by LC-MS/MS to prevent blockage of the instrument. Matrix effects occur where phospholipids and salts affect the efficiency of ionisation of the parent/daughter compounds (or quantifier and qualifier), which can be reduced by sample treatment strategies. However, this can affect sensitivity, increase cost, and delay turnaround time, as well as other drawbacks [49].

LC-MS/MS can also be affected by ion suppression—another type of matrix effect where components of the specimen coelute—preventing effective ionisation of the analyte and thus, interfering with detection and quantitation [23]. An example of this is antibiotic interference, piperacillin, in urinary cortisol assays [23]. However, the chromatography step helps reduce the risk of ion suppression and the use of isotope-labelled internal standards can detect the ion suppression. The internal standards should be selected carefully to avoid spuriously low calculated concentrations of the analyte and to ensure that they are not interfered with by the metabolites of the analyte generated during measurement [23,49]. 

In summary, even if the cortisol method is very specific, patient preparation remains key. This includes the cessation of interfering drugs (Table 1) or the collection of medication history, and any interfering drugs that cannot be stopped should be discussed with the laboratory. It is well known that glucocorticoids should be stopped before the measurement of cortisol. Despite this, Keevil reports that, in the first year of introducing routine LC-MS/MS for cortisol quantification, 33% of specimens received in the clinical laboratory contained detectable prednisolone [49]. 

#### 3.2.2. HPLC

Separation of analytes by HPLC increases analytical specificity. A variety of detection techniques for cortisol and cortisone analysis can then be selected, e.g., with fluorescence, diode array, and ultraviolet [34]. However, the lower limit of detection of these detection techniques are often too high; therefore, they are no longer widely used and have been surpassed by mass spectrometry [34]. Fenofibrate has been reported to interfere in HPLC and HPLC-MS/MS single transition methods in urine (see Table 1) [51]. Carbamazepine can also be falsely detected in HPLC cortisol assays (as well as speeding up dexamethasone metabolism), spuriously increasing the measured concentration [52]. 

### 3.3. Specimen Type and Timing

Cortisol can be analysed in a range of body fluids, and it is again vital that patient preparation and timing are controlled [1]. There is diurnal variation of cortisol release, with nadir shortly after going to sleep (approximately midnight) and a peak in the morning. In people who are normally awake during the day, the concentration starts to rise from 3 am [9]. Loss of the diurnal variation can be observed in hypercortisolism, which, in clinical practice, is the assessment of the maximum suppression (nadir concentration), e.g., midnight cortisol [1]. However, not all patients with Cushing syndrome will invariably have loss of the circadian rhythm [53]. Random specimens are therefore of no use, and sample collection has to either be timed or be part of a dynamic function test. An exception is the immediate post-operative period [54].

#### 3.3.1. Plasma/Serum (Midnight)

Midnight plasma total cortisol analysis is a second line test for Cushing syndrome as it is impractical and stressful to admit a person to fall asleep and then be bled in the hospital [1]. Plasma analysis of total cortisol is affected by conditions affecting CBG concentration (see discussion above), with free hormone analysis not being widely available nor superior (as are estimations) to salivary analysis [55,56]. 

Specificity of midnight cortisol (both plasma and salivary) is reduced due to the loss of diurnal variation of cortisol in many other conditions besides Cushing syndrome [1]. For example, people living with depression or obesity have all been shown to be at risk of having raised midnight salivary cortisol concentrations, and the nadir may also be affected by work and sleep patterns, e.g., shift workers [45,57,58].

#### 3.3.2. Saliva (Midnight)

Salivary cortisol improves diagnostic accuracy of cortisol assessment as it is both ‘free’ (not bound to CBG) and provides the facility of doing midnight specimen collections, without having the stress and resource implications of admitting a person to the hospital for blood tests. These features mean that midnight salivary cortisol assessment is recommended as one of the initial tests that can be used in assessing possible Cushing syndrome [1,34,44,59]. It may also be a suitable tool for assessment during pregnancy and of remission of Cushing syndrome post surgical treatment [44,60,61,62].

Two consecutive collections are recommended to minimise the risk of contamination (from blood and steroids on hands). However, others have instead suggested that a single measurement is sufficient, and elevated results should be followed up with a dexamethasone suppression test for improved sensitivity and specificity [1,34,42,63,64]. Although midnight is the time recommended in guidelines, high sensitivity and specificity has been shown from specimens taken at 22:00 and 23:00 h [33].

Use of Salivette^®^ (Sarstedt AG & Co. KG, Nümbrecht, Germany), a cotton swab chewed for 1–2 min, may be better correlated with serum cortisol concentrations compared with passive drooling [65]. Specimens are stably refrigerated for at least 7 days, if not 3 months, with only minor instability at room temperature, allowing specimens to be delivered to the laboratory at ambient temperature [66,67]. Due to the viscous nature of saliva, freezing the specimens prior to analysis and/or centrifugation can precipitate compounds, allowing the specimen to be sampled and analysed more easily [66]. 

Liquorice, which inhibits 11β-HSD2 and therefore increases cortisol (and decreases cortisone) concentration, should not be eaten for 4 days prior to the collection of saliva [37]. Grapefruit juice, amongst many other compounds, should also be avoided prior to the test (please see metabolism section above) [33,38,68]. 

Contamination of the Salivette^®^ by hands to which topical hydrocortisone had been applied will significantly increase the cortisol concentration, as will ingestion of oral hydrocortisone, due to contamination of the mouth [35,37]. A cortisol:cortisone ratio of >1 can be used to diagnose contamination of a salivary specimen [33]. Administration of intravenous hydrocortisone does not interfere up to 12 h post dose [35]. Smoking increases salivary cortisol concentration and the cortisol:cortisone ratio, leading to the recommendation to avoid smoking on the day of collection [33,69]. 

If the saliva sample is contaminated with blood, as identified by visual inspection, then a repeat sample should be considered as the blood cortisol will significantly contribute to the measured concentration [37]. Therefore, salivary cortisol may be inappropriate for people with oral or dental diseases, such as ulcers, and it is preferable to avoid eating and tooth brushing for at least an hour prior to collecting the sample. The mouth should also be rinsed well with water at least 30 min before saliva collection [34,66]. 

The matrix of saliva itself may cross-react with immunoassays and produce unreliable results [70]. Salivary cortisol reference intervals may not be widely applicable. For example, age, hypertension, and diabetes may increase midnight salivary cortisol concentrations [71]. 

#### 3.3.3. Urine (24 h)

The comparative diagnostic accuracy of cortisol measured in urine recommends it to be used as one of the possible first line tests to confirm hypercortisolaemia [1,30]. Urinary free cortisol (UFC) can also be used to assess remission, for example, surgical success [1,61,62]. As the cortisol is ‘free’, it is useful for the assessment of pregnant individuals (and others, see Section 2) in whom the CBG is elevated, which would cause a higher total cortisol result independently of the biologically active free fraction [1]. The 24 h duration of collection also provides an integrated assessment of concentration over and above a single blood or salivary specimen. However, the diagnostic efficiency may not be better than saliva in some cohorts, but this may be due to analytical cross-reactivity [30,72]. 

Conversely, in order to appear in the urine, the plasma concentration of free cortisol must exceed the binding capacity of plasma. Therefore, if a person has a consistent but mild elevation of plasma total cortisol, the UFC may be spuriously negative [30]. There is poor correlation between UFC and severity of clinical phenotype [73].

Mass spectrometry detection of cortisol metabolites in urine may be more sensitive, but it is not routinely available (although clinical services increasingly exist) [14]. Immunoassays are at risk of over-quantifying the urinary cortisol concentration, likely due to cross-reactivity. Therefore, equivalent diagnostic efficiency cut-offs will need to be method specific [74,75]. Two collections are recommended as variability is high [1,76].

Obesity and metabolic syndrome can increase UFC, as will any true increase in cortisol, irrespective of the cause [1,77]. High fluid intake can increase UFC significantly (if taking in excess of 5 L a day and using the upper limit of normal as cut off), as can a liberal salt intake [78,79]. The cross over with non-Cushing syndrome and high UFC is high; therefore, UFC concentrations up to four-fold the quoted upper limit of normal are required for diagnostic accuracy, whilst maintaining adequate specificity [1].

False negative results can occur with mild or cyclical Cushing syndrome [1,43]. Free cortisol excretion is also affected by renal function [80,81]. As renal function deteriorates, UFC falls in concentration. A threshold of calculated creatinine clearance of <60 mL/min is recommended, below which UFC should no longer be used to confirm hypercortisolism, due to the risk of a false negative [1,80]. Intraindividual variability is high, with approximately 50% variation when cortisol is analysed by HPLC [76].

More general issues with 24 h urine collections include the inconvenience for patients, risk of over or under collection, or specimens never being returned [82,83]. Taking four random urine specimens and drying them has been shown to be equivalent to a 24 h collection and may be a more practical option for some patients and some health care economies [84]. There is a risk of accidental or deliberate contamination in self-collected urine specimens, e.g., by ingestion or direct addition to urine of glucocorticoids [27,29]. 

Grapefruit juice, likely due to inhibition of 11β-HSD2, increases urinary cortisol excretion [68]. Depending on cut-offs used, the performance of saliva and urine is similar for cortisol detection [85].

#### 3.3.4. Plasma/Serum (Early Morning)

Combining early morning and later night total cortisol measurements to demonstrate the loss of diurnal variation (and thus detect hypercortisolism) has been studied, but it is not currently standard practice [86]. Sampling in the morning, e.g., 8 am, is therefore mainly reserved for monitoring management success [61,62,87]. For example, an early morning plasma cortisol concentration <55 nmol/L is associated with a high chance of remission, post pituitary surgery, for Cushing disease [88]. 

#### 3.3.5. Plasma/Serum (Free)

Cortisol measurements in blood are usually total cholesterol (i.e., free cholesterol and protein bound cholesterol). Serum free cortisol analysis is not widely available. Preparation of specimens is labour intensive, with the risk of interference from zinc sulphate reagent in immunoassays [89]. Calculations are limited by our current inability to accurately quantify CBG in a clinically useful way (significant range in tertiary structure and affinity for cortisol) [89]. The role of serum free cortisol is therefore limited, particularly when LC-MS/MS steroid profiles can make up for some of the limitations of immunoassay plasma cortisol, and for those patients with medically treated Cushing syndrome (see below).

#### 3.3.6. Hair

Hair, due to slow growth rate, can be used to demonstrate long term exposure to glucocorticoids [90,91]. This specimen type is yet to be widely used in clinical practice due to a range of issues, including lack of standardised collection and storage procedures, reference materials and reference intervals, and uncertainty regarding interferents and physiology [92]. Hair cortisone may improve diagnostic efficiency; however, hair cortisol and cortisone may miss cases in mild Cushing syndrome phenotypes [93,94]. 

## 4. ACTH 

Only once hypercortisolism has been confirmed should ACTH measurement be considered, particularly as random concentrations are difficult to interpret unless there is an established abnormal cortisol concentration [1,62]. If ACTH secretion is suppressed, e.g., <10 pg/mL (ng/L) (levels vary between assays), an adrenal source (if exogenous excluded) of cortisol is indicated. If ACTH concentration is inappropriately high, e.g., above 20 pg/mL, or an intermediate concentration, then an ACTH-dependent cause is indicated [62]. ACTH measurement may also be considered to assess remission post-surgery [61].

ACTH is present in very low concentrations; compare 20 ng/L of ACTH with a low albumin concentration of 20 g/L or 20,000,000,000 ng/L. ACTH is small at 4.5 kDa, but not as small as cortisol. ACTH also has a very short half-life (between 6–7 min in vivo); therefore, specimens need careful handling to avoid spurious results [95]. Due to instability, specimens for ACTH analysis must be kept at room temperature, be centrifuged, and separated from the cells within 2 h [96,97]. Both haemolysis and excessive tube anticoagulant (EDTA) can also cause spuriously low results [97,98]. 

### 4.1. ACTH Analysis

#### 4.1.1. Techniques

ACTH is widely measured by sandwich immunoassay design. It is therefore at risk of interference from antibodies or other substances, such as a negative bias with biotin (see discussion above) [24,99]. Caution should be exercised with assays with relatively high lower limits of detection. The assay must be able to accurately report very low concentrations, at least <5 ng/L (pg/mL), otherwise true suppression can be missed as the performance of the assay is not good enough to distinguish concentrations around the lower limit of detection, which is too high in the first place [27,29,99]. There is no international standard or reference method for ACTH, preventing comparison of results from one assay to another.

#### 4.1.2. Performance

Ectodermal tissues can produce ACTH. Therefore, if there is no pituitary neuroendocrine tumour secreting inappropriate ACTH, then another possible tumour of ectodermal origin should be investigated [100]. The most common tumour type is lung, either carcinoid or small cell [100]. However, ACTH is not elevated in all people with ectopic ACTH secretion, e.g., approximately 72% in one study [100]. Other pro-opiomelanocortin-derived (POMC) peptides can be co-secreted. However, any tumour that secretes ACTH will have expression of the POMC gene, preventing the use of POMC peptides to localise tumours to the pituitary.

## 5. Other Steroid Hormones

ACTH stimulates the production of adrenal androgens in the zona reticularis in the adrenal glands. Therefore, women presenting with hirsutism or virilisation, and other signs of Cushing syndrome, may have an ACTH-dependent Cushing syndrome or carcinoma of the ovary or adrenal [13]. Congenital enzyme defects may also be a possible cause in young women, e.g., non-classical congenital adrenal hyperplasia (CAH). Measurement of other steroid hormones, in addition to cortisol, may help indicate which of these pathologies is the cause. 

### 5.1. Adrenal Androgens in Women

#### 5.1.1. Dehydroepiandrosterone-Sulfate

Dehydroepiandrosterone is synthesised in the adrenal cortex and circulates in a sulphated form, dehydroepiandrosterone-sulfate (DHEAS). DHEAS can be used to indicate the activity of the adrenal glands. For example, in autonomous cortisol production by an adrenal adenoma, DHEAS concentration is suppressed, but performance in diagnostic strategies has not been robust enough to be included in guidelines [1,101,102]. Carafone et al. suggested that a combination of high DHEAS and ACTH can exclude autonomous cortisol secretion in adrenal adenoma cohorts, preventing the necessity to conduct any further tests [102]. In women, when virilisation may be due to an ovarian tumour, DHEAS can be useful as it is primarily made in the adrenal glands, whereas androstenedione originates from both the adrenals and ovaries in practically equal amounts. Therefore, a raised androstenedione is not specific to either organ, but an elevated DHEAS may decrease the chance of an ovarian source [13]. DHEAS will be raised when hypercortisolism is ACTH-dependent [13,93].

#### 5.1.2. Other Female Androgens

Although rare, the differential for Cushing syndrome in young women includes non-classical CAH. This autosomal recessive genetic condition is detected by synthetic ACTH-stimulated 17-hydroxyprogesterone [13]. Testosterone is not helpful in differentiating a potential adrenal tumour from an ovarian tumour in a woman presenting with hirsutism, as the circulating concentration is equally from both endocrine glands and peripheral conversion of androstenedione [13].

### 5.2. Steroid Profile

The measurement of urine steroid metabolites is more complex but provides a wealth of information. Initially, profiles analysed by gas chromatography, and either mass spectrometry or a flame ionisation detector, were shown to detect Cushing syndrome and separate it from mild hypercortisolism seen in adrenal incidentalomas [14]. Subsequently, LC-MS/MS techniques have been developed that are robust and suitable for discriminating adrenal carcinoma from adrenal incidentaloma or adenoma [103,104,105]. Undeniably, the steroid metabolome can provide useful information in diagnosing both hypercortisolism and its cause, and the developing technology is improving feasibility and method performance [49,106,107,108].

### 5.3. Salivary Cortisone

Salivary cortisone (see above discussion of liquorice in saliva section) has been shown to be more sensitive and just as specific as salivary cortisol in the assessment of Cushing syndrome [33,109]. The sum of cortisol and cortisone concentrations in saliva may perform better than a late-night salivary cortisol concentration alone, but not better than cortisone alone [33]. Measurement is via ultra-HPLC with LC-MS/MS; therefore, it is not routinely available in all laboratories and the superiority over measuring salivary cortisol via immunoassay remains controversial [33,110]. Cortisone immunoassays do exist, but they are not in routine use, with LC-MS/MS methods reporting both cortisol and cortisone in one test when available [34]. Salivary cortisone is also free from interference from contamination by oral and intravenous hydrocortisone, so it can be used to indicate the efficacy of replacement therapy [35]. The lack of interference by blood and liquorice (as compared to salivary cortisol) makes salivary cortisone a potentially more robust screening test for Cushing syndrome [37]. 

## 6. Dynamic Function Tests

A range of dynamic function testing protocols exist to improve the diagnostic efficiency of cortisol tests. Suppression protocols, such as the dexamethasone suppression test (DST), can help confirm the initial diagnosis, followed by a range of anatomical and stimulation tests to identify the aetiology [1]. A full discussion is outside of the scope of this review, but a brief summary of the impact on laboratory testing is provided below.

### 6.1. Dexamethasone Tests

Reduced negative feedback by glucocorticoids in autonomous hormone production is another first line method to detect the presence of pathological endogenous glucocorticoid production. Relying mainly upon plasma cortisol concentrations, these suppression tests are affected by factors that affect plasma cortisol quantification (i.e., an increase in CBG from hormone therapy can produce a misleadingly high total cortisol result). Dexamethasone does not cross-react in most cortisol assays and is a potent glucocorticoid, which should suppress cortisol production. However, it is debated if the degree of suppression is not that specific to the underlying condition [111]. Additionally, there are significant differences in dexamethasone concentration reached in plasma, likely due to variation in absorption and metabolism between individuals [1]. 

In addition, many drugs affect metabolism [1]. Rifampacin and many other medications have also been shown to speed up dexamethasone metabolism, preventing the expected suppression of the hormone axis [112,113]. Simultaneous measurement of dexamethasone can be used to reduce the risk of spuriously positive results and, therefore, could be considered in those with unsuppressed cortisol concentrations. Unfortunately, dexamethasone assays are not currently widely available [114,115]. Liver and renal failure may be associated with reduction in the clearance of dexamethasone and a risk of false negative results [1].

#### 6.1.1. 1 mg Overnight Dexamethasone Suppression Test

The 1 mg overnight dexamethasone suppression test (ODST) protocol is associated with high diagnostic accuracy as a first line test to confirm the presence of hypercortisolism, except in pregnancy, possible cyclical Cushing syndrome, and epilepsy (if antiepileptics enhance dexamethasone suppression) [1]. Cut-offs will need to be appropriate for local cortisol method, but stringent thresholds are recommended [1].

#### 6.1.2. 2 mg/Day (Low Dose) 48 h Dexamethasone Suppression Test

A lower dose (0.5 mg four times a day), longer treatment protocol with dexamethasone is an alternative to the 1 mg ODST in the assessment of people with possible Cushing syndrome. This longer treatment protocol can have a higher specificity; therefore, it is commonly preferred by some practitioners [1]. It is particularly useful in possible pseudo-Cushing cases with high UFC, but, in general, diagnostic accuracy is not as high as the ODST and UFC [1]. It may be preferable to the ODST in women on the contraceptive pill [116].

#### 6.1.3. High Dose Dexamethasone Suppression Test

A high does dexamethasone suppression test (HDST) of either 8 mg overnight or 2 mg (or 8 mg) four times a day for 48 (or 24) hours may be used when the 1 mg ODST shows non-suppressed cortisol concentration [117,118]. This may help distinguish between ectopic and pituitary ACTH sources, but it is controversial and not recommended to establish the diagnosis of Cushing syndrome [1,118,119,120].

#### 6.1.4. Dexamethasone-CRH Test

The dexamethasone-CRH test is considered a second line test to help confirm the presence of hypercortisolism [1,121]. However, measurement of dexamethasone is recommended, the cortisol assay needs to be accurate at very low concentrations, and studies remain limited; therefore, caution should be exercised with diagnostic thresholds [1,122].

#### 6.1.5. Dexamethasone Analysis in DST Protocol

Consideration of dexamethasone analysis can prevent spurious results due to accelerated or reduced dexamethasone metabolism (endogenous or drug-induced) [1]. Analysis is not widely available and it may be that this resource is better limited to cases where there is a high suspicion of spurious results, as it is also not universally useful [115]. 

### 6.2. CRH Stimulation with ACTH and Cortisol Quantification

People with Cushing disease have a more pronounced increase in ACTH and cortisol concentration after CRH administration than those with ectopic ACTH, but it is not recommended to establish a diagnosis of Cushing syndrome [1,118]. Performance is superior to the 8 mg overnight dexamethasone suppression test and is recommended over it when a pituitary adenoma has not been identified on imaging; however, CRH is difficult to obtain [26,123]. 

### 6.3. Desmopressin (DDAVP) Test

The desmopressin (DDAVP) test is a potentially useful test once hypercortisolism is confirmed [1,26,118]. Although there may be a role in pseudo-Cushing syndrome (with similar performance to the dexamethasone-CRH test), it is recommended for distinguishing Cushing disease from ectopic ACTH, particularly in non-invasive test protocols which combine dynamic function tests and imaging [26,124,125]. Desmopressin may also be a more easily available and cheaper alternative to CRH in invasive testing protocols [126].

### 6.4. Anatomical

There are a range of anatomical tests that can be used to help identify the source of the autonomous cortisol or ACTH production [1]. Although a detailed review is outside the scope of this paper, it is worth noting that these tests also have pitfalls, including significant overlap in results, e.g., in mild cases versus those without Cushing syndrome [29]. Practical challenges may revolve around resource, canulation of the vessels, invasiveness of test, and accurate labelling of specimens. Combining imaging and dynamic function tests may be a suitable alternative in most cases [126,127].

## 7. Other Tests

One of the main clinical concerns in screening for Cushing syndrome is missing a significant life-limiting condition, such as carcinoma. There can be a temptation to image immediately or measure various tumour markers. Caution must be taken to avoid further inappropriate testing and management. For example, although an ACTH producing ovarian cancer can cause hypercortisolaemia, it is comparatively a rare cause, and CA-125 is non-specific for carcinoma, as it can be negative or elevated in other conditions (e.g., menstruation, renal failure, and lung and pancreatic cancers). Therefore, CA-125 has no role for screening for the cause in a case of Cushing syndrome [128,129]. 

### 7.1. Neuroendocrine Markers

Neuroendocrine tumours (NETs) are very rare, but related biomarkers may have a limited role in cases of confirmed ACTH-dependent hypercortisolism [130] and have therefore been included here for completeness. However, caution must always be taken when using tumour markers as they are non-specific to carcinoma; therefore, clinical suspicion as to the site and type of tumour should be high before use.

#### 7.1.1. Chromogranin A

Chromogranin A (CgA) is stored in secretory granules (with other granin family molecules, e.g., chromogranin B) in most neuroendocrine cells and neurons [131]. It can be released as a full-length molecule or various active fragments [131]. In suspected cases of ectopic-ACTH, a raised chromogranin A can be both supportive of a NET cause and subsequently used as a tumour marker [3,130]. 

Proton pump inhibitors can increase the CgA concentration significantly; therefore, they need to be stopped at least seven days prior to testing. Histamine receptor 2 antagonists require stopping at least 24 h before CgA sample collection [132]. CgA is also elevated in many other conditions, such as non-NET cancers, renal failure, and after glucocorticoid use or a meal, limiting its use as a screening tool; it is also higher in the afternoon than in the morning [131,132,133]. Positive analytical interference has been described in the presence of haemolysis or fibrin in the specimen and due to antibody interference [131]. 

Due to the presence of fragments and a variety of post-translational modifications, immunoassay sensitivity differs depending on the manufacturer, due to which epitope the antibodies detect [133,134]. Both antibody type and assay design affect the results obtained [135]. Sample stability varies widely between individuals which may be due to polymerisation, complex formation, folding, and degradation in the specimen [134]. Therefore, the current use of CgA is somewhat limited due to biological, pathophysiological, and analytical factors and requires specific cautions when collecting, e.g., specimen kept on ice and rapid centrifugation. 

#### 7.1.2. Serotonin (5-Hydroxyindole Acetic Acid)

Cushing syndrome has been described in cases of a type of NET, carcinoid; carcinoid tumours are the most common cause of cyclical Cushing syndrome due to ectopic ACTH secretion [3,130,136]. Carcinoid tumours produce biogenic amines, of which serotonin is used as a screening tool. The metabolite of serotonin, 5-hydroxyindole acetic acid (5HIAA), is widely measured in 24 h urine collections. However, specificity and sensitivity are limited [3,130,137,138]. Platelet serotonin is not affected by diet, unlike urine 5HIAA, and is more sensitive but not widely available [137,138]. Again, the use in Cushing screening is therefore negligible, but in suspected/proven carcinoid tumours, it may be a useful marker to monitor treatment success. 

#### 7.1.3. Calcitonin

In cases of ACTH-dependent Cushing syndrome, calcitonin may be elevated in ectopic ACTH cases, but never in pituitary causes [3,139]. Importantly, however, calcitonin is not specific to the tumour type or location, and it can also be elevated in carcinoid tumours, medullary thyroid cancer, small cell lung cancer, and various other types and locations [139]. This therefore limits the test to being mainly useful in monitoring only.

#### 7.1.4. Copeptin

Copeptin—a cleavage peptide from ADH (vasopressin) production—has been proposed as a marker of glucocorticoid activity, as glucocorticoids suppress ADH synthesis [140]. Copeptin is a cleavage molecule produced in equimolar amounts to ADH from their precursor, preprovasopressin. However, ADH is difficult to accurately measure as it is very unstable and produced at very low concentrations in a pulsatile fashion [141]. Measured by immunoassay, copeptin assays are at risk of heterophilic antibody interference, and concentration is affected by osmolality [142]. The performance has yet to be shown as a useful adjunct in the management of Cushing syndrome, nor is the test widely available [140]. 

### 7.2. Electrolytes, Acid Base, Physiological

#### 7.2.1. Hypokalaemic Alkalosis

Due to cross-reactivity at mineralocorticoid receptors, hypokalaemia and alkalosis can be observed in hypercortisolism cases. Although occurring in adrenal adenoma, hypokalaemia is common in ectopic ACTH cases and has been reported to affect 71% of cases in one cohort [100,143]. If a very severe and rapid onset of Cushing syndrome is seen with hypokalaemic alkalosis, then ectopic ACTH becomes the most likely endogenous cause [119].

#### 7.2.2. Metabolome

For a discussion of the steroid metabolome, please see the section above on steroid profiles. Exposure to excess glucocorticoids can result in complications, such as diabetes mellitus and dyslipidaemia. Assessment of the metabolome using ultra-high-performance liquid chromatography-tandem mass spectrometry has been shown to discriminate hypercortisolaemic patient specimens versus eucocortisolaemic [144]. It is possible that this type of technique, including molecular (see below), may have value in borderline cases where neither clinical nor biochemical findings can confirm or refute the diagnosis of hypercortisolaemia, but it has yet to be used outside a research setting. 

### 7.3. Molecular Genetic Techniques

Although molecular techniques are not yet widely available nor routinely used in Cushing syndrome, the rapid reduction in cost and the improving specificity of these technologies is changing the research field. Examples include studies on DNA methylation, which regulates gene expression, e.g., FKBP5 (FK506-binding protein 51) mRNA, and protein expression is directly stimulated by glucocorticoid receptor agonism [145]. Although endogenous and exogenous glucocorticoids will increase FKBP5 concentration, it may help to confirm hypercortisolism in those with borderline cortisol concentrations, as variation in GR physiology may explain phenotype variation (i.e., why some people may present symptomatic earlier at more subtle cortisol concentration elevations than others). In summary, it is proposed that the change in methylome may demonstrate tissue exposure to glucocorticoids better than isolated hormones [145,146].

Methylation patterns of leucocyte DNA have also been shown to discriminate Cushing syndrome from other causes of endocrine hypertension, making the idea of a methylome hypertension screen an attractive proposition [147]. Other examples include using the gene signatures of adrenal tumours to assess prognosis and sensitivity to medications [148].

### 7.4. Others

#### Anti-Müllerian Hormone and PCOS

Anti-Müllerian hormone (AMH), produced in the granulosa cells of the ovary, is raised in polycystic ovarian syndrome (PCOS), so it may be helpful in discriminating mild hypercortisolism and PCOS, which can be difficult clinically [13]. Luteinising hormone (LH) and sex hormone binding globulin (SHBG) both increase in PCOS, whereas they fall in concentration in Cushing syndrome [13].

## 8. Testing in Medically Treated Cushing Syndrome

Medical therapies can be used to treat hypercortisolaemia, whilst awaiting operations, or if no procedure is available or tolerable to the patient. It is vital to recognise how these drugs affect the assessment of cortisol status so that doses can be titrated to cause remission and to minimise the risk of adrenal insufficiency.

### 8.1. Steroid Synthesis Inhibition

Adrenal steroid synthesis can be inhibited by ketoconazole, mitotane, osilodrostat, and metyrapone, among others [26]. However, these agents result in structurally similar metabolites to cortisol, such as 11-deoxycortisol, which risks over treatment of spurious hypercortisolism or missing hypoadrenalism due to over treatment [25,26]. It is therefore vital to assess cortisol with an assay free from interference in the presence of steroidogenesis inhibition, and the method of choice would be LC-MS/MS methods validated to be free from interference [25,144]. However, protocols involving immunoassay with urine and saliva have been reported [149]. 

Mitotane also increases CBG concentration [150]. Therefore, immunoassay may overestimate the total cortisol concentration further, increasing the risk of over treatment.

### 8.2. Central Inhibition

Somatostatin receptor agonism with pasireotide, or dopamine receptor agonism with cabergoline, can reduce cortisol secretion of adrenal adenomas via reduction in POMC transcription. Theoretically, these therapies should be free from interference in any assessment of either cortisol or ACTH, and there are no known cases of interference to our knowledge.

### 8.3. Glucocorticoid Receptor Antagonism

Mifepristone can be used to help control hyperglycaemia in Cushing disease as it blocks cortisol from binding to glucocorticoid receptors. However, it is not widely used and can present significant risks [151]. One particular issue is that neither ACTH nor cortisol concentration can be used to reliably assess the degree of hyper- or hypocortisolism (as either increase or remain the same); therefore, clinical assessment must be relied upon [151]. Due to cortisol cross-reactivity with renal mineralocorticoid receptors, significant hypokalaemia—which may be exacerbated by an increase in cortisol concentration due to an increase in ACTH because of a reduction in negative feedback—is a significant side effect [151]. This apparent mineralocorticoid excess can mask significant hypocortisolaemia in mifepristone use [151]. 

## 9. Conclusions

Although there are significant differences between the performance of different analytical techniques, specimen types, and dynamic function protocols, all have their relative merits and limitations. Poor analytical specificity may increase diagnostic sensitivity, and there is always a balance to be made with cost and laboratory resources. Combining thorough clinical assessment with an understanding of test performance should allow most cases of hypercortisolaemia to be accurately diagnosed and monitored without the use of resource intensive analysis. However, it is without doubt that improvements in technology and knowledge will continue to improve diagnostic efficiency, and it may be time to consider the role of urine steroid profiles and salivary cortisone in guidelines. 

## Figures and Tables

**Figure 1 diagnostics-13-01415-f001:**
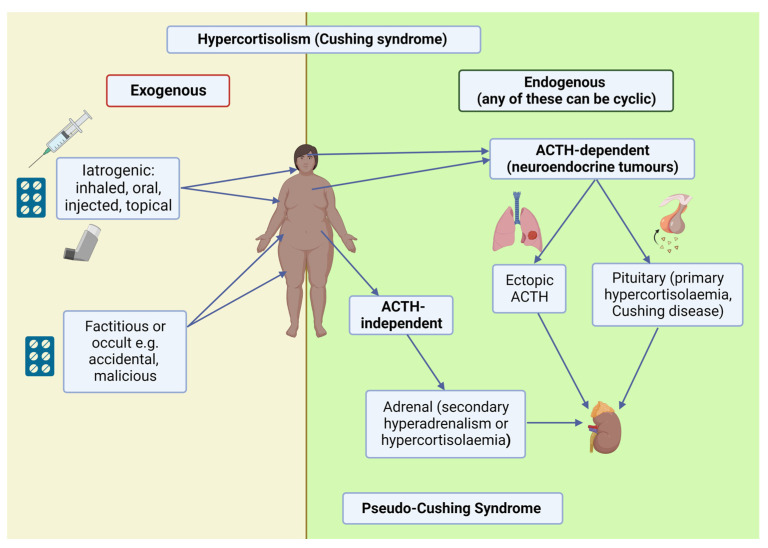
Causes and nomenclature of hypercortisolism (Cushing syndrome) in humans. Once exogenous sources of glucocorticoids have been excluded, then endogenous hypercortisolism is divided into ACTH dependent and ACTH independent causes. Pseudo-Cushing syndrome is associated with signs, symptoms, and hypercortisolaemia, but is a consequence of other clinical conditions, such as obesity or depression. Created with Biorender.com [1].

**Figure 2 diagnostics-13-01415-f002:**
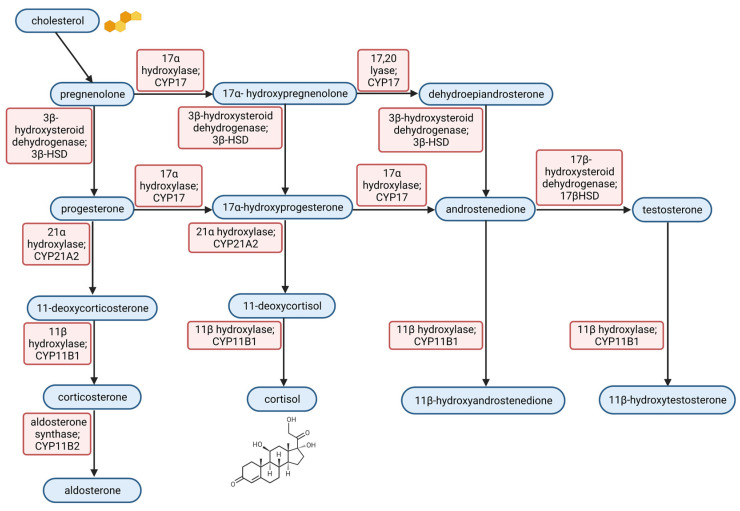
Steroid synthesis pathway. Steroid hormones are mostly synthesised in the adrenal cortex, with some sex steroid steps occurring in the gonads (not demonstrated here) from cholesterol. The small changes and multiple steps result in a wide range of structurally similar end hormones and precursors, which can cross-react with both assays and in vivo receptors. Created with Biorender.com [13,14,15].

**Figure 3 diagnostics-13-01415-f003:**
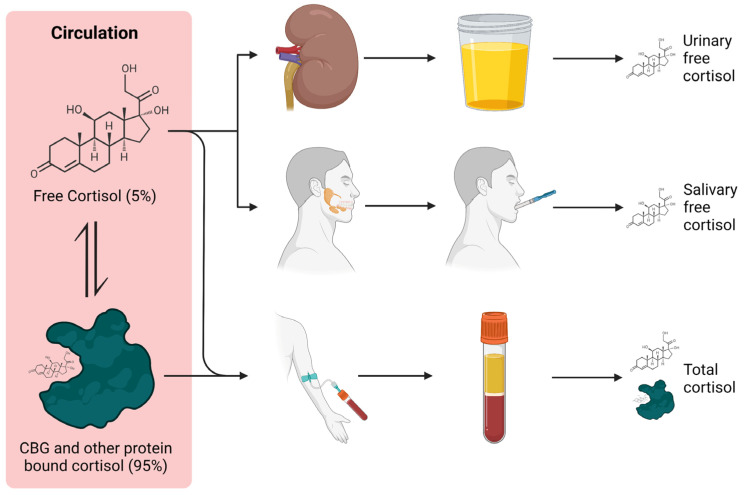
Cortisol fractions and specimens. The majority of circulating cortisol is protein bound; therefore, blood sampling primarily provides an assessment of total cortisol concentration. Free cortisol can be detected in urine and saliva specimens. Created with Biorender.com.

**Figure 5 diagnostics-13-01415-f005:**
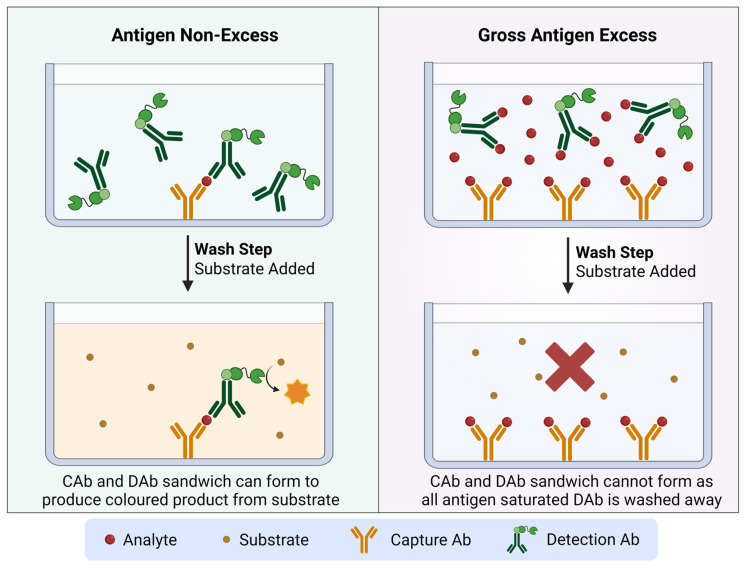
Mechanism of antigen excess interference—also called the hook/prozone effect—in one step sandwich immunoassay systems. Detection of antigen occurs when a sandwich is formed by the binding of the capture antibody (CAb) and the detection antibody (DAb) to the antigen. When antigen is in excess, sandwiches are not formed, as all binding epitopes are saturated by antigen, hence the DAb lost in the wash step. Created with Biorender.com.

**Table 1 diagnostics-13-01415-t001:** Examples of drugs causing false results in cortisol analysis [1,23,24,25,26,27,28].

False Positive		False Negative	
Drug	Mechanism; Assay	Drug	Mechanism; Assay
Carbamazepine	Induction of CYP 3A4 *, accelerates metabolism of dexamethasone; any	Aprepitant	Inhibition of CYP 3A4 *, inhibition of dexamethasone metabolism; any
Ethosuximide		Cimetidine	
Phenobarbital		Diltiazem	
Phenytoin		Fosaprepitant	
Pioglitazone		Fluoxetine	
Primidone		Itraconazole	
Rifampin		Ritonavir	
Rifapentine			
Estrogens	Increase CBG; plasma total	Biotin	Competes for binding; streptavidin-based IA
Mitotane			
Carbamazepine	Overlaps cortisol peak; UFC by HPLC or single transition HPLC MS/MS	Piperacillin	Ion suppression; MS UFC
Fenofibrate	Interferes with HPLC peak; UFC HPLC		
Synthetic glucocorticoids	Cross-reactivity; IA		
Carbenoxolone	Inhibition of 11β-HSD2; UFC and salivary; any		
Biotin	Competes for binding; streptavidin-based IA		
Steroid synthesis inhibitors	Cross-reactivity of steroid precursors; any particularly IA		
Mifespristone	Result correct but tissue activity blocked, cortisol analysis should not be performed; any		

* see The Flockhart Cytochrome P450 Drug-Drug Interaction Table for further information [28]. Key: 11β-HSD2, 11beta-hydroxysteroid dehydrogenase type 2; HPLC, high performance liquid chromatography; IA, immunoassay; MS, mass spectrometry; UFC, urinary free cortisol.

## Data Availability

Not applicable.

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
