# Peer review of "Pitfalls in the Diagnosis and Management of Hypercortisolism (Cushing Syndrome) in Humans; A Review of the Laboratory Medicine Perspective"

_diagnostics, 2023, doi:10.3390/diagnostics13081415_

Round 1
Reviewer 1 Report
The paper of Kade C Flowers1 and Kate E Shipman is a review mainly focusing on the issues of the technical tools used for the diagnosis of Cushing’s syndrome.
The paper is comprehensive and well-written. Nevertheless, there are several crucial mistakes and imprecisions.
L114 : what about the role of estrogens given in post-menopausal women ?
L 144 : can the exposure to phthalates (plasticisers), organotins (found in paint), alkylphenols and perfluorinated substances have an influence on biological testing ?
L 192: unclear ? immunoassays have low sensitivity to diagnose adrenal insufficiency ??????
L321 : some papers claim that the stability of salivary cortisol is longer than 7 days in the fridge. Please mention that these can be send to the lab at ambient temperature.
L367 : disagree : UFC is within the normal range in the majority of obese patients or patients with metabolic syndrome
L 451-452: I do not understand the last sentence of the paragraph.
L 457 spelling error for “hirsutism”
L457 : ectopic ACTH secreting tumors responsible for paraneoplastic CS also induce increased secretion of DHEA and, consequently, hirsutism !
L 527 : unclear. Which point is debated?
L557 : some teams do not recommend to perform the HDDST for the differentiation between ectopics and Cushing’s disease. This is a point of high controversy that should be mentioned (see ref 128)
L560: one major issue with the Dex-CRH test is the lack of consensus on the biological criterion for positivity
L 574 : in all large series published the performance of the CRH test is unambiguously better than that of high dose 8 mg test !!! look back to the litt and also see the recents reviews and recommendations. the reference quoted (118) is totally inappropriate !!!!!!
L574 : Note that CRH is no longer available universally
L 578 Desmopressin test : the test may be useful ! the references quoted are totally inappropriate. If you look to the latest guidelines ref 25/ 2021 rather than cited ref 1 /2008 (some opinions based on studies have changed between 2008 and 2021 !!!) you will read that the desmopressin test can be used for the differential diagnosis between ectopic ACTH secretion and Cushing’s disease. There are also studies showing that the desmopressin test may help to define groups at high vs low risk of recurrence following pituitary surgery for Cushing’s disease (C Eur J Endocrinol. 2020; Eur J Endocrinol. 2020).
Author Response
Please see attachment, thank you.

Reviewer 2 Report
Thank you for this comprehensive review. I have a few suggestions:
· I would use the spelling “Cushing’s syndrome”.
· First sentence: Hypercortisolism is not the same as Cushing’s syndrome. Please rewrite this sentence.
· In Figure 2: The 11-oxygenated androgens are missing and should be added as they are clinically important.
· “For an excellent review, please see Krone et al 2010 [30]” – this is too judgmental.
· “The diagnostic accuracy of cortisol measured in urine is high, allowing it to be used as one of the possible first line tests to confirm hypercortisolaemia [1, 29].” – This sentence is very debatable!
· Regarding CRH-Test: “Performance is similar to the 8 mg overnight dexamethasone suppression test [118]”. – This is not exactly true, please do another literature research.
· 7.1. Neuroendocrine Markers: The entire section seems to be far beyond the topic – or at least the clinical practice. Ectopic Cushing’s syndrome is so rare and measuring neuroendocrine markers is very, very rare in clinical practice. I would recommend deleting this section.
Round 2
Reviewer 1 Report
adequately revised for me